# Gradient Fan-in Asymmetry: The Structural Cause of Layer Redundancy in Deep Transformers

## Abstract

Deep Transformers are composed of uniformly stacked residual blocks, yet their deepest layers often add little value. Prevailing explanations attribute this to small gradients, treating a symptom rather than the cause. We identify *Gradient Fan-in Asymmetry* as the structural driver of redundancy. In Pre-LayerNorm residual stacks, the gradient at a layer is the sum of an identity path and all downstream functional paths, producing a gradient fan-in that decays linearly with depth (and quadratically under deep supervision), yielding rich signals early and sparse for later layers. Across Transformers and ResNets, accumulated training gradients follow the theoretical fan-in and predict post hoc layer importance. Two causal interventions isolate structure as the bottleneck: equalizing per-layer gradient norms does not restore late-layer value, whereas increasing downstream path counts via parameter-shared repetition restores and elevates their impact. Building on this mechanism, we propose CascadeFlow Pruning, which removes layers using accumulated training gradients and outperforms standard heuristics without expensive post hoc analysis. We also introduce CascadeFormer, which tapers width with depth to match the natural information flow, achieving comparable perplexity to a uniform baseline at the same training budget while reducing latency by 8.6% and increasing throughput by 9.4%.

## 1 Introduction

The uniform scaling of transformer blocks Vaswani et al. (2017), simply repeating identical layers to create deeper models, has been the driving principle behind the success of Large Language Models (Radford et al.; Brown et al., 2020; Touvron et al., 2023). However, this architectural homogeneity masks a significant functional asymmetry. For instance, evaluating a pretrained LLaMA model on WikiText, deeper layers exhibit high representational similarity, a key indicator for redundancy (Gromov et al., 2024) (Figure 1a). This asymmetry is even more pronounced in architectures like LayerSkip (Elhoushi et al., 2024), which skips later layers by exiting the network early, revealing their sharply declining functional contribution (Figure 1b).

The conventional explanation for this phenomenon points to attenuated gradients in deeper layers Li et al. (2025). This observation, while correct, mistakes a symptom for the cause. We argue the root issue is not the gradient's magnitude, but its compositional diversity, a structural bottleneck we term *Gradient Fan-in Asymmetry (GFA)*. The residual connections (He et al., 2016b) that enable deep training transform the network into an implicit ensemble of many paths of varying lengths (Veit et al., 2016), which creates a fundamental imbalance during backpropagation.

In this work, we argue that layer redundancy is a direct consequence of a training dynamic we term **Gradient Fan-in Asymmetry (GFA)**. This is not an attenuation of mere magnitude, but rather of compositional diversity. Due to the path ensemble structure, shallow layers receive gradient from all subsequent functional blocks. This creates a cascade where their updates are compositionally rich, while the deepest layers, aggregating from few blocks, receive a structurally simple and information-poor gradient.

We validate the GFA hypothesis and demonstrate its utility through a sequence of empirical arguments. First, we establish a strong correlation between per-layer gradient norms $\bar{g}_i$ and even-

tual functional importance $\Delta\mathcal{M}_i$. We then move beyond correlation with two causal interventions: one ablative, showing that artificially amplifying late-layer gradient magnitude fails to restore their importance, and one constructive, showing that structurally increasing their path counts via layer repetition does restore it. These interventions confirm the bottleneck is structural. Finally, we translate this causal insight into two practical applications: **CascadeFlow Pruning (CFP)**, an efficient method leveraging accumulated training gradients to outperform standard pruning heuristics, and the **CascadeFormer**, an architecture that tapers width with depth to align model capacity with the natural flow of compositional gradient diversity, improving inference efficiency at fixed training FLOPs.

This work reframes gradient magnitude not as a cause to be fixed, but as a proxy for a structural information imbalance. Our contributions are:

- **We identify and validate Gradient Fan-in Asymmetry as the causal mechanism** for layer redundancy. We provide strong causal evidence through two complementary interventions: an ablative test (artificially equalizing gradient norms fails to restore importance) and a constructive one (structurally increasing path counts via layer repetition succeeds). This isolates the bottleneck to the gradient's compositional complexity, not its raw magnitude.

- **We introduce CascadeFlow Pruning (CFP)**, an efficient method that leverages accumulated training gradients as a high-fidelity proxy for structural importance to prune layers, outperforming standard heuristics without requiring expensive post-hoc analysis.

- **We design the CascadeFormer**, an architecture that internalizes the GFA principle. By tapering network width with depth to match the natural flow of compositional gradient diversity, it reduces latency and increases throughput over a uniform baseline with equal training FLOPs and comparable perplexity.

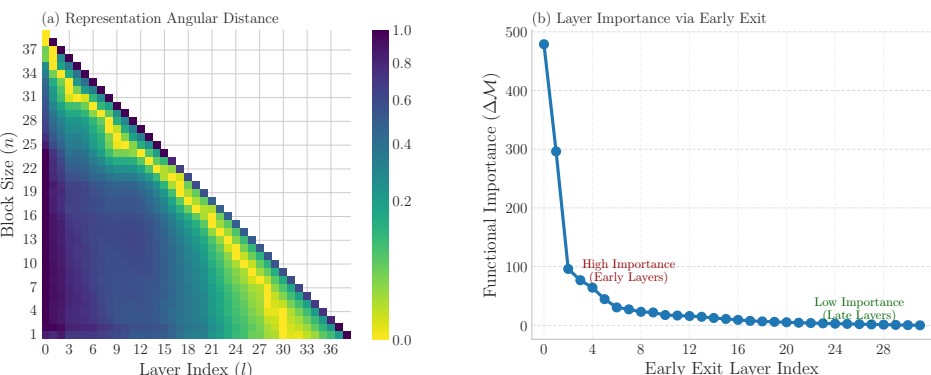

Figure 1: **Deeper Transformer layers show diminishing contributions.** (a) In a LLaMA 13B model, representational similarity across layers increases with depth, signaling growing redundancy. (b) A LayerSkip LLaMA 8B makes the consequence explicit: layer importance, measured by Functional Importance ($\Delta\mathcal{M}$) upon its removal, is concentrated in the initial layers while the functional value of later layers collapses.

## 2 RELATED WORK

**Layer Redundancy for Model Efficiency.** Deep networks exhibit layer redundancy that can be exploited for compression and faster inference (Gromov et al., 2024; Sun et al., 2025; Chen et al., 2025). Structured pruning removes entire blocks with minor loss (Chen et al., 2023; Frantar & Alistarh, 2023; Ma et al., 2023; Xia et al., 2024; Kim et al., 2024; He et al., 2024; Sun et al., 2024), while training-time methods like LayerDrop (Fan et al., 2020) and early-exit/skip mechanisms (Elhoushi et al., 2024; Xin et al., 2020; Liu et al., 2020; Zhao et al., 2025; Men et al., 2025) allow dynamic redundancy management.These works document *that* redundancy exists and how to use it; they do not explain *why* it arises.

**The Architectural Roots of Redundancy.** Residual networks (He et al., 2016b;a) can be viewed as implicit ensembles of shorter paths Veit et al. (2016), a property inherited by Transformers with Pre-LayerNorm architectures (Xiong et al., 2020; Touvron et al., 2023). This structure often causes deeper layers to contribute minimally, leading to redundancy (Takase et al., 2023; Sun et al., 2025; Li et al., 2025). A complementary line of work shows how architectural and normalization choices regulate gradient propagation across depth (Wang et al., 2022; Shleifer et al., 2021; Li et al., 2024). Our method is instead applicable to the Pre-LN setting, and building on these results, we argue and test that depth-wise allocation of gradient signal is structurally induced and exploiting this phenomenon to optimize for more efficient architecture depth wise and width wise.

**Quantifying Layer Importance.** Identifying redundant layers requires a reliable importance metric. Magnitude and first/second-order criteria provide strong baselines (Han et al., 2015; 2016; Molchanov et al., 2017; Lee et al., 2019; Frantar & Alistarh, 2023). Other methods rely on measuring the effects of redundancy: output-similarity (e.g, cosine between adjacent layers), correlates high similarity with low importance (Gromov et al., 2024; Yang et al., 2024; Jiang et al., 2025; Chen et al., 2025; Song et al.), while perturbation-based metrics such as ($\Delta$PPL) upon single layer removal directly quantify its functional contribution (Kim et al., 2024). Gradient-based signals have been used as local surrogates for importance, but typically in heuristic form (e.g., saliency Smilkov et al. (2017); Selvaraju et al. (2020), Taylor criteria (Yang et al., 2023; Ma et al., 2023)). We instead posit and test a *causal mechanism*: gradient dynamics are not just a proxy; they *drive* the final functional hierarchy via a compositional gradient asymmetry.

## 3 GRADIENT FAN-IN ASYMMETRY

**The Phenomenon.** We identify Gradient Fan-in Asymmetry (GFA), a structural imbalance in the composition of gradient signals within deep residual architectures. This asymmetry arises because shallow layers receive gradients aggregated across numerous downstream computational paths, while deep layers receive them from a progressively smaller set. This disparity in fan-in directly governs the *compositional diversity* of the resulting gradient. Consequently, deep-layer gradients are compositionally simple and thus information-poor, leading to less effective weight updates. Crucially, this is a structural limitation, not a magnitude problem; optimizers that only rescale gradients cannot correct this underlying informational deficit.

**GFA in Residual Networks.** We define a Pre-LN Transformer as $x_{l+1} = x_l + F_l(x_l)$, where $F_l$ represents the blocks complete transformation, including LayerNorm and sublayers. The gradient at its input, $g_l \equiv \partial\mathcal{L}/\partial x_l$, unrolls into a cumulative sum over all downstream blocks:

$$g_l = g_N + \sum_{k=l}^{N-1} J_k^T g_{k+1}, \tag{1}$$

where $J_k$ is the Jacobian of the $k$-th block's transformation. This structure means the gradient at layer $l$ aggregates signals from an identity path and all subsequent functional paths. We term the number of these aggregated signals the gradient fan-in, $\phi_l$, which decreases linearly with depth (visualized by the solid paths in Fig. 2. A formal counting rule is provided in Appx. B.1.

**Amplification via Deep Supervision.** Architectures employing deep supervision, such as LayerSkip, amplify GFA. By introducing auxiliary loss heads at intermediate layers, they create new gradient hierarchies that backpropagate to shallower layers. This transforms the linear fan-in disparity into a quadratic one (illustrated by the additional dotted paths in Fig. 2), severely concentrating gradient information in the shallowest layers. The full derivation in Appx. B.1 shows this formally. This exacerbates, rather than solves, the structural imbalance.

**Analysis and Prediction.** It is crucial to distinguish our fan-in proxy from the $2^{N-l}$ combinatorial paths that arise from a full expansion. Our proxy measures the number of distinct signal channels aggregated at a layer, not their information quality (e.g., orthogonality), which remains an empirical question. This leads to our central thesis: *gradient norm is a symptom, not the cause*. A small gradient norm in a deep layer reflects its limited access to compositional information. An optimizer like AdamW Loshchilov & Hutter (2018) can rescale this gradient, but it cannot invent the rich,

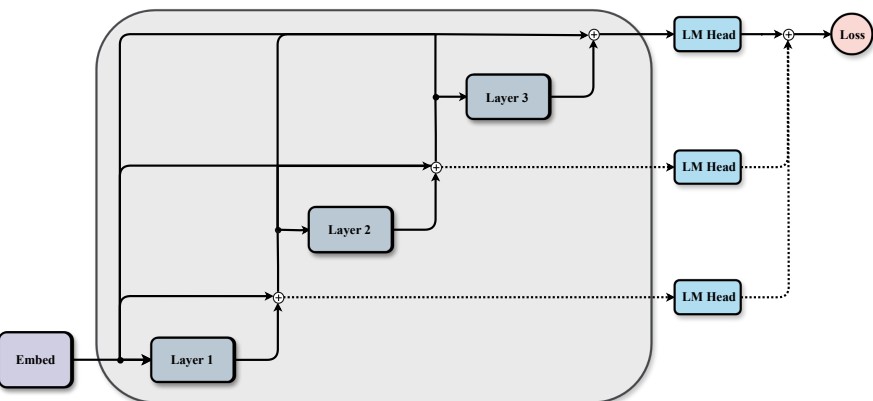

Figure 2: **Gradient Fan-in Asymmetry arises from a structural imbalance in gradient paths.** The unwrapped view shows that the gradient at any layer $l$ is a sum over signals from an identity path and all subsequent functional paths (Eq. 1). The number of contributing functional paths (solid lines) decreases linearly with depth, creating a compositional asymmetry. Deep supervision (dotted lines) exacerbates this imbalance by introducing new gradient hierarchies from auxiliary losses, which transforms the fan-in disparity from linear to quadratic.

compositional information that is structurally absent. From this principle, we derive a testable prediction: any architectural change that structurally increases a layer's downstream gradient fan-in will increase its functional importance. We test this directly with virtual depth and use the principle to design our methods.

## 4 EMPIRICAL VALIDATION AND APPLICATIONS

We test Gradient Fan-in Asymmetry in three stages. We first show that gradient flow is structurally skewed toward early layers and that this training signal predicts the final functional hierarchy. We then perform two interventions that separate magnitude from information content and isolate structure as the cause. Finally, we translate the mechanism into two applications, CascadeFlow Pruning and CascadeFormer, that improve efficiency at fixed training cost.

### 4.1 SETUP

**Models and architectures**   We evaluate three residual families. For language modeling we train a sixteen layer, approximately 1.2B parameter, Llama base Transformer (Touvron et al., 2023) referred to as Vanilla and a LayerSkip variant (Elhoushi et al., 2024). For vision we train a ResNet-50 (He et al., 2016b). To encode the GFA prior we modify the Vanilla architecture to create CascadeFormer which tapers width with depth to align capacity with the decay in compositional gradient diversity. All models are trained from scratch.

**Datasets and tasks**   Language models are trained on a seven billion token subset of Dolma (Soldaini et al., 2024) for next token prediction. ResNet-50 is trained on ImageNet-1K (Deng et al., 2009). Training hyperparameters and optimizer settings are in Appendix A. For our primary architectural comparison, we train the proposed CascadeFormer$_{A_2}$, the full baseline, and a 15-layer Baseline, using three different random seeds. In contrast, all other models mentioned in our analysis (16-layer Vanilla, LayerSkip, ResNet-50) were initialized with a single seeds.

### 4.2 QUANTIFYING GRADIENT FLOW AND FUNCTIONAL IMPORTANCE

To empirically test our hypothesis, we require metrics that can directly link training dynamics with the final functional hierarchy of the model.

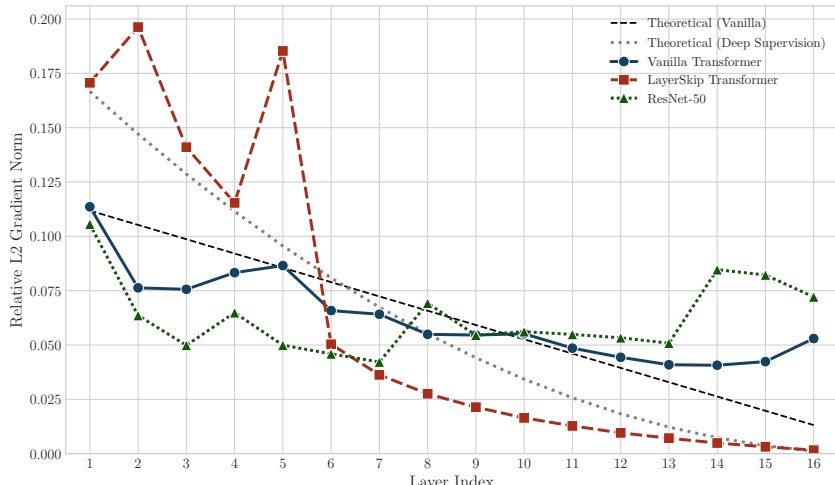

Figure 3: **Gradient flow is inherently front loaded in deep residual architectures.** The average L2 gradient norm per layer in a Vanilla Transformer follows a linear decay, closely tracking its theoretical fan-in (black, dashed). In contrast, LayerSkip's deep supervision mechanism induces a quadratic decay, a behavior accurately modeled by its own theoretical curve (gray, dotted). ResNet-50 also exhibits a characteristic front-loaded decay, confirming that gradient distribution is a direct.

**Accumulated Gradient Share** ($\bar{g}_i$). To capture a layer's overall contribution during training, we accumulate the L2 norm of the gradients with respect to its parameters, $\theta_i$, over $T$ training steps. This accumulated value is then normalized by the total sum from all $N$ layers to yield the relative gradient share, $\bar{g}_i$:

$$\bar{g}_i = \frac{\sum_{t=1}^{T} \|\nabla_{\theta_i}\mathcal{L}_t\|_2}{\sum_{j=1}^{N}\sum_{t=1}^{T}\|\nabla_{\theta_j}\mathcal{L}_t\|_2}. \tag{2}$$

This metric serves as a direct, data-driven proxy for the structural information flow predicted by GFA.

**Functional Importance** ($\Delta\mathcal{M}_i$). We quantify a layer's functional importance by measuring the performance degradation when its contribution is removed. This is achieved by ablating layer $i$, bypassing its computational block while preserving the residual path to the subsequent layer. The functional importance, $\Delta\mathcal{M}_i$ is the absolute degradation in the task metric $M$ resulting from this ablation. For language models, this is the increase in perplexity ($\Delta$PPL), and for vision models, the change in top-1 accuracy ($\Delta$Acc). A larger $\Delta\mathcal{M}_i$ signifies greater functional importance.

### 4.3 CORRELATIONAL EVIDENCE ACROSS ARCHITECTURES

**Structural fan in aligns with gradient flow** GFA predicts that the downstream gradient fan in decreases with depth, inducing a front loaded gradient distribution. Figure 3 illustrates this empirical pattern across three architectures alongside their theoretical fan in decay. For the vanilla network, this decay is linear, while for LayerSkip it is quadratic. We observe that the theoretical fan in and the empirical gradient norm $\bar{g}_i$ share the same monotonic ordering. Complete derivations and path counts are provided in Appendix B.1.

**Gradient flow forecasts the functional hierarchy** We correlate each layer's $\bar{g}_i$ with its ablation based importance $\Delta\mathcal{M}_i$. Figure 4 shows a strong positive Spearman correlation for the Vanilla Transformer with $\rho = 0.62$ and $p = 0.02$ and for ResNet-50 with $\rho = 0.83$ and $p < 0.01$. The relationship tightens in LayerSkip with $\rho = 0.99$ and $p < 0.01$ where supervision accentuates early layer fan in. This links the structural gradient skew during training to the final functional hierarchy, establishing gradient as a reliable proxy for functional importance.

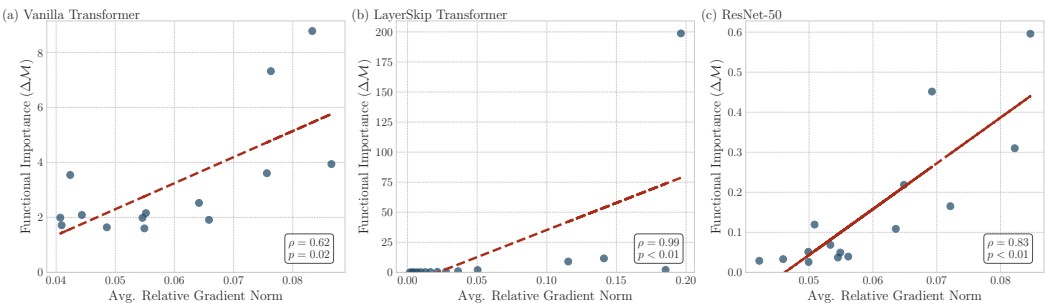

Figure 4: **Training gradient flow predicts final layer importance.** The accumulated gradient share $\bar{g}_i$ during training shows a strong Spearman correlation with post-hoc functional importance $\Delta \mathcal{M}_i$. The correlation is significant in the Vanilla Transformer with $\rho = 0.62$ and in ResNet-50 with $\rho = 0.83$, and is near perfect in LayerSkip with $\rho = 0.99$.

## 4.4 CAUSAL INTERVENTIONS ISOLATE STRUCTURE AS THE BOTTLENECK

Correlation does not imply causation. To disentangle the roles of gradient magnitude and structure, we conduct two interventions designed to directly test the GFA hypothesis.

**Equalizing magnitude does not restore importance** We first test the alternative hypothesis: that small gradient magnitude is the direct cause of redundancy. During training, we insert a hook that rescales per-layer gradients to have an equal L2 norm. To accommodate this artificial amplification, we proportionally scaled the gradient clip norm, deriving the factor from the ratio of maximum observed norms between the hooked and standard models, details shown in Appendix A.2.2. If magnitude were the causal factor, this should rescue the importance of later layers. The result, shown in Figure 5 (top), is the opposite. This logarithmic comparison against a vanilla model, with an inset detailing the validation gradient distribution not only fails to restore importance but actively harms the contribution of deep layers. Amplifying an information-poor signal does not make it complex; it merely makes the simple signal louder, potentially destabilizing learning.

**Increasing path counts restores importance** Next, we directly test the structural component of GFA against a vanilla 8 layer reference. We engineer an increase in the downstream gradient paths for deep layers by repeating the final four layers of an 8-layer model with shared parameters, which increases a layer's virtual depth and its gradient fan-in without adding parameters. Specifically, we repeat the last four layers with a pattern of two, three, three, and four repeats respectively, a pattern designed to increase the fan-in with depth. This modification dramatically alters the structural fan-in for the deepest layers (detailed in Appx. B.1), for instance changing the counts for layers 5 through 8 from [5, 4, 3, 2] to [27, 33, 24, 20]. GFA predicts these layers should become more functionally important.

Figure 5 (bottom) validates this prediction. The inset shows that gradients in the repeated deep layers increase substantially, even exceeding those of early layers. Consequently, their functional importance rises, diminishing the relative contribution of the initial four layers. These modified deep layers, now recipients of a more compositionally complex gradient, become more critical than the untouched shallow layers. Together, these interventions provide strong causal evidence that functional hierarchy is governed by the structural flow of information, for which gradient magnitude is a correlated proxy

## 4.5 APPLICATIONS INFORMED BY GFA

Having validated GFA as the causal mechanism, we now demonstrate its practical utility for model efficiency through a superior pruning method and a novel, GFA-aware architecture.

**CascadeFlow Pruning uses training dynamics** We prune layers using the accumulated gradient share, $\bar{g}_i$, gathered directly during training. Our method, CascadeFlow Pruning (CFP), uses the L2

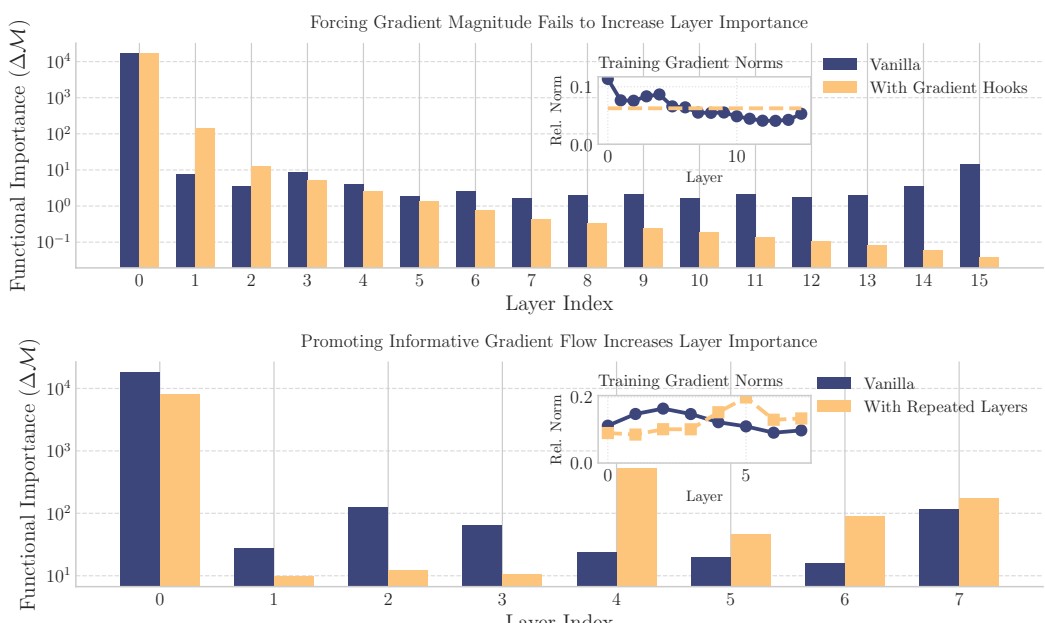

Figure 5: **Causal tests confirm that gradient structure dictates importance.** Top equalizing L2 gradient norms across layers does not rescue and reduces deep layer importance. Bottom increasing downstream path counts by parameter shared repetition restores and elevates late layer importance. The logarithmic y-axis is necessary to visualize the wide dynamic range. The crucial observation is how each layer's importance changes relative to itso baseline.

norm of these gradients (Equation 2) as a proxy for a layer's functional importance, ranking them accordingly. This approach eliminates the need for expensive post hoc computation required by alternative heuristics like hidden state similarity Gromov et al. (2025), Taylor based methods Ma et al. (2023), or parameter magnitude pruning Han et al. (2015). CFP then removes the lowest ranked layers, irrespective of their original consecutive block structure.

EVALUATION We evaluated pruning strategies on the Dolma 2.6M token holdout set and evaluation set of HellaSwag benchmark. We report perplexity (PPL) on Dolma. On HellaSwag, we adopt the zero-shot protocol from OLMo OLMo et al. (2024), ranking candidates by their relative conditional log-probability. While this zero-shot approach does not measure post-fine-tuning adaptability, it provides a direct and efficient benchmark for quantifying performance degradation as a function of pruning.

Table 1 summarizes the results. Similarity method becomes be competitive only under the most aggressive pruning, but typically at a higher perplexity cost. In contrast, Taylor and Magnitude methods are unstable. Their reliance on a per-layer forward pass yields importance rankings that fluctuate across seeds, leading to misrankings and sharp performance degradation. Our CFP avoids this sensitivity, producing stable rankings and a more graceful degradation profile

IMPLEMENTATION VIA LAYER PASSTHROUGH Pruning a Transformer layer in our framework requires no architectural modification: we implement it by a simple *passthrough* in the forward pass since skipped layers are simply treated as identity functions. Specifically, during inference the model iterates over the stack of decoder layers. For each index $i$, if $i$ belongs to the pruned set $\mathcal{I}_{\text{prune}}$, we skip the forward call to the corresponding block and directly forward the input hidden state to the next layer:

$$h_{i+1} = h_i, \qquad \text{if } i \in \mathcal{I}_{\text{prune}},$$

This design choice allows CFP to be integrated into existing Transformer codebases (e.g., `LLaMA`-style decoders) with only a few lines of modification.

| $k$ | Metric | CFP (Ours) | | Sim | | Taylor | | Magnitude | |
|---|---|---|---|---|---|---|---|---|---|
| *Baseline (k=0)* | | PPL=17.94 $\pm$ 0.00, Acc.=0.39 $\pm$ 0.00 | | | | | | | |
| 1 | PPL $\downarrow$ | **19.848 $\pm$** | **0.082** | 21.945 $\pm$ | 0.462 | 24.700 $\pm$ | 0.559 | 24.361 $\pm$ | 2.357 |
| | Acc. $\uparrow$ | 0.381 $\pm$ | 0.002 | 0.381 $\pm$ | 0.002 | **0.384 $\pm$** | **0.001** | 0.362 $\pm$ | 0.013 |
| 2 | PPL $\downarrow$ | **23.226 $\pm$** | **0.106** | 28.480 $\pm$ | 0.331 | 127.744 $\pm$ | 15.148 | 41.875 $\pm$ | 4.822 |
| | Acc. $\uparrow$ | **0.372 $\pm$** | **0.001** | 0.366 $\pm$ | 0.001 | 0.369 $\pm$ | 0.003 | 0.325 $\pm$ | 0.006 |
| 4 | PPL $\downarrow$ | **59.790 $\pm$** | **1.840** | 59.790 $\pm$ | 1.840 | 4715.072 $\pm$ | 4166.454 | 332.174 $\pm$ | 216.582 |
| | Acc. $\uparrow$ | 0.334 $\pm$ | 0.004 | 0.334 $\pm$ | 0.004 | **0.336 $\pm$** | **0.002** | 0.285 $\pm$ | 0.010 |
| 6 | PPL $\downarrow$ | **167.006 $\pm$** | **9.205** | 180.862 $\pm$ | 27.304 | 1193.159 $\pm$ | 547.709 | 3099.530 $\pm$ | 1708.037 |
| | Acc. $\uparrow$ | 0.299 $\pm$ | 0.004 | 0.304 $\pm$ | 0.003 | **0.305 $\pm$** | **0.002** | 0.269 $\pm$ | 0.002 |
| 8 | PPL $\downarrow$ | **911.748 $\pm$ 55.001** | | 911.748 $\pm$ 55.001 | | 1403.434 $\pm$ | 434.894 | 1237.212 $\pm$ | 132.241 |
| | Acc. $\uparrow$ | **0.285 $\pm$** | **0.001** | 0.285 $\pm$ | 0.001 | 0.278 $\pm$ | 0.006 | 0.264 $\pm$ | 0.001 |

Table 1: **CFP demonstrates superior performance and stability under aggressive layer pruning.** We evaluate CFP (Ours) against standard pruning heuristics by removing an increasing number of layers ($k$). CFP consistently achieves the lowest perplexity (PPL) and maintains competitive downstream accuracy (Acc), particularly at higher sparsities.

**CascadeFormer.** Our second application internalizes the GFA principle directly into the model's architecture. Because compositional gradient diversity decays with depth, uniform capacity allocation is inherently inefficient. We therefore designed the **CascadeFormer**, an architecture that tapers model width to align its capacity with this information flow. For a model with $N$ layers, indexed $l \in \{0, \ldots, N-1\}$, we apply tapering rules to either the attention, FFN sublayers or both.

ATTENTION TAPERING. We reduce the number of attention heads, and thus the attention dimension $d_{\text{attn}}(l)$, in discrete steps governed by, $d_{\text{attn}}(l) = d_{\text{attn},0} - S_d \cdot \lfloor l/F_d \rfloor$, where $d_{\text{attn},0}$ is the initial dimension, $S_d$ is the dimensional reduction per step, and $F_d$ controls the frequency of the reduction.

FFN TAPERING. We reduce the FFN's inner dimension $d_{\text{ffn}}(l)$ linearly with depth according to the rule, $d_{\text{ffn}}(l) = d_{\text{ffn},0} - S_f \cdot l$, where $d_{\text{ffn},0}$ is the initial dimension and $S_f$ controls the steepness of the linear taper.

VARIANT CONFIGURATIONS. We define six CascadeFormer variants based on these rules, categorized by low (subscript 1) and high (subscript 2) tapering intensity. The specific hyperparameters for each variant are detailed in Table 4. The combined variants (C) apply both attention and FFN tapering schemes simultaneously. To ensure a fair comparison, we also trained six baseline models whose computational cost was scaled linearly by reducing their layer count.

Table 2 quantifies the performance and efficiency of our GFA-informed architectures. Our CascadeFormer-$A_2$ was designed with a training FLOP budget equivalent to the Vanilla-15L baseline. It achieves comparable perplexity to the baseline ($17.84 \pm 0.02$ vs $17.84 \pm 0.03$) while reducing inference latency by 8.6% and increasing throughput by 9.4%.

To further understand the design space, we explored additional variants applying the tapering principle to FFN layers ($F_1$, $F_2$) and in combination ($C_1$, $C_2$). While all GFA-informed models are competitive, the superior performance of the attention-tapered variants ($A_1$, $A_2$) suggests that the primary structural bottleneck identified by GFA resides within the self-attention mechanism, making it the most effective target for tapering.

**Latency Measurement Protocol.** All inference metrics were measured on a single A100 GPU. To ensure fair comparison, we used a consistent batch size, context length, and generation length for all models. We leveraged `torch.compile` with the `mode='max-autotune'` and `fullgraph=True` options to minimizing implementation-specific overhead and accurately reflecting the inherent hardware-friendliness of each architecture. We report the median of 100 timed runs after a warmup phase of 10 steps to measure accurate execution time.

| | Core Metrics | | Hardware Efficiency | | | |
|---|---|---|---|---|---|---|
| **Model** | **PPL** $\downarrow$ | **Params** (B) | **Util.** (TFLOP/s) $\uparrow$ | **Cost** (TFLOPs) | **Latency** (ms) $\downarrow$ | **Throughput** (tok/s) $\uparrow$ |
| *Uniform Baselines* | | | | | | |
| Vanilla-16L | 17.69 | 1.28 | $53.87 \pm 0.30$ | 4.82 | $89.47 \pm 0.50$ | $22{,}890 \pm 129$ |
| **Vanilla-15L** | 17.84 | 1.21 | $54.23 \pm 0.34$ | 4.54 | $83.81 \pm 0.53$ | $24{,}439 \pm 153$ |
| Vanilla-14L | 18.00 | 1.15 | $54.68 \pm 0.38$ | 4.27 | $78.09 \pm 0.54$ | $26{,}227 \pm 181$ |
| Vanilla-13L | 18.17 | 1.08 | $55.18 \pm 0.39$ | 4.00 | $72.40 \pm 0.52$ | $28{,}287 \pm 201$ |
| Vanilla-12L | 18.82 | 1.01 | $55.74 \pm 0.34$ | 3.72 | $66.75 \pm 0.40$ | $30{,}683 \pm 186$ |
| *CascadeFormer (Ours)* | | | | | | |
| CascadeFormer-$A_1$ | 17.79 | 1.25 | $55.39 \pm 0.37$ | 4.72 | $85.17 \pm 0.57$ | $24{,}048 \pm 161$ |
| **CascadeFormer-$A_2$** | 17.84 | 1.22 | $59.77 \pm 0.38$ | 4.58 | $76.62 \pm 0.48$ | $26{,}731 \pm 169$ |
| CascadeFormer-$F_1$ | 17.88 | 1.19 | $51.16 \pm 0.27$ | 4.43 | $86.65 \pm 0.47$ | $23{,}635 \pm 127$ |
| CascadeFormer-$F_2$ | 18.10 | 1.09 | $49.92 \pm 0.24$ | 4.05 | $81.06 \pm 0.39$ | $25{,}266 \pm 123$ |
| CascadeFormer-$C_1$ | 17.94 | 1.16 | $53.00 \pm 0.35$ | 4.33 | $81.71 \pm 0.54$ | $25{,}066 \pm 164$ |
| CascadeFormer-$C_2$ | 18.30 | 1.03 | $53.88 \pm 0.49$ | 3.81 | $70.65 \pm 0.66$ | $28{,}990 \pm 266$ |

Table 2: **CascadeFormer produce a superior balance of performance and hardware efficiency.** Our CascadeFormer-$A_2$ model, which tapers its attention mechanism according to GFA principles, outperforms its Vanilla-15L baseline that was trained with an equivalent computational budget. It achieves similar perplexity to baseline but is substantially faster, reducing inference latency by 8.6% while increasing throughput by 9.4%.

## 5 DISCUSSION AND CONCLUSION

This work reframes layer redundancy in residual networks not as a failure of optimization, but as a predictable outcome of their structure. We identified and causally validated Gradient Fan-in Asymmetry as the root mechanism. This insight is not merely diagnostic; it is generative. It led directly to CascadeFlow Pruning, a more effective pruning method, and CascadeFormer, an efficient architecture that aligns its capacity with the asymmetric flow of information.

**Design Tension and Future Directions.** Our findings present a fundamental design tension. One path is to embrace the asymmetry, as CascadeFormer does, leading to intentionally heterogeneous architectures that allocate resources where learning dynamics can best use them. A second path is to counteract GFA, aiming to force uniform functional contribution. This second path is complicated by evidence from architectures like LayerSkip. By imposing deep supervision, these models create an extreme gradient hierarchy that forces shallow layers to become functionally self-sufficient. This serves as an unintentional proof of concept: shallow layers possess a vast latent capacity that standard end-to-end training, governed by a gentler GFA decay, fails to fully exploit. This suggests the true limitation may not be layer capacity but the training dynamic itself. Such a pursuit would require new architectural components or training schemes that can inject compositional diversity into deep layer gradients, perhaps through novel long range information pathways or regularization techniques. Whether a uniform contribution is achievable, or even desirable, is a critical open question.

## LIMITATIONS.

Our analysis frames Gradient Fan-in Asymmetry in terms of path quantity, using fan-in as a proxy for compositional diversity. A crucial next step is to analyze the quality of these gradient signals, their effective rank, orthogonality, and information content, which may provide a more complete picture. Our empirical validation is conducted on models up to 1.2B parameters; while the GFA principle is architectural, its precise dynamics at the 100B+ parameter scale remain an open empirical question. Finally, our proposed pruning method, CFP, requires access to training-time gradients, making it inapplicable for post-hoc pruning of pre-trained, closed-source models.

## REPRODUCIBILITY STATEMENT

To ensure the reproducibility of our findings, we provide comprehensive details on our methodology, data, and computational environment. **Code:** The complete codebase to reproduce all experiments, including model training, evaluation, and figure generation, will be made publicly available on GitHub upon publication. **Environment:** The hardware and software stacks for training (TPU v4) and evaluation (NVIDIA A100/A6000), including all library versions, are documented in Appendix A.1. **Data:** All experiments utilize public datasets. Language models were trained on the publicly available pre-tokenized Dolma dataset from on Huggingface. Vision models were trained on the standard ILSVRC 2012 ImageNet-1k dataset. **Methodology:** Full architectural specifications for all models, including our novel CascadeFormer, are in Appendix A.2 and A.2.1. The precise methodology for our causal intervention experiment, which demonstrates that amplifying late-layer gradients is insufficient for improving their importance, is detailed in Appendix A.2.2. **Hyperparameters:** All training and evaluation hyperparameters are enumerated in Appendix A.3. All experiments can be reproduced with fixed random seed of 324709 for model initialization and data loading.

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

## USE OF LARGE LANGUAGE MODELS

In adherence to ICLR guidelines, we disclose the use of Large Language Models (LLMs) as assistive tools in the preparation of this manuscript. Our use was focused on two areas: manuscript preparation and literature discovery.

**Manuscript Preparation**  We utilized an LLM for copy-editing tasks, including correcting grammar, refining prose for clarity, and ensuring stylistic consistency. The core intellectual contributions, including the research questions, methodology, analysis, and conclusions, were conceived and formulated exclusively by the authors.

**Literature Discovery**  An LLM was employed as a preliminary tool to broaden our literature search. Its function was to identify potentially relevant papers and suggest alternative search keywords based on our initial research scope. Every paper cited in this work was subsequently read, critically evaluated, and integrated into our analysis by the authors to confirm its relevance and correctness.

## A  EXPERIMENT DETAILS

This section provides comprehensive details on our experimental setup to ensure full reproducibility. We detail the model architectures, training hyperparameters, and the specific implementation of our proposed CascadeFormer variants as well as layer-wise pruning heuristic details.

### A.1  COMPUTATIONAL RESOURCES AND SOFTWARE

#### A.1.1  TRAINING ENVIRONMENT

All model training was conducted on Google Cloud Platform (GCP) using a 128-core TPU v4 Pod slice. Our training stack utilized PyTorch `2.7.0` with the corresponding `torch_xla==2.7.0` library.

To distribute training, we employed the GSPMD (Xu et al. (2021)) partitioner. We utilized a 1D sharding strategy where model parameters, gradients, and optimizer states were fully sharded across the data-parallel dimension. This approach efficiently managed memory while scaling computation across all 128 TPU cores. All training was performed using a `float32` data type.

#### A.1.2  EVALUATION ENVIRONMENT

All evaluations were performed on a single GPU, using either an NVIDIA A100-80GB or an NVIDIA A6000-48GB. The evaluation framework was PyTorch `2.7.0`. All experiments were conducted in `float32` precision to mititgate any possibility of precision error.

#### A.1.3  IMPLEMENTATION DETAILS

Our model architecture is a modification of the Llama implementation from the Hugging Face `transformers==4.51.3` library. This same codebase served as the basis for our layer-dropping ablation studies. All experimental results were logged using the Weights & Biases (`wandb`) platform, and figures included in this paper were generated with `matplotlib`.

### A.2  MODEL CONFIGURATIONS

The core of our investigation involves three distinct architectures to test the generalizability of our claims. Their primary configurations are summarized in Table 3.

#### A.2.1  CASCADEFORMER ARCHITECTURE DETAILS

The CascadeFormer internalizes the GFA principle into its structure. While standard Transformers allocate uniform capacity to all layers, our architecture recognizes that compositional information

| Parameter | Vanilla Transformer | LayerSkip Transformer | ResNet-50 |
|---|---|---|---|
| Base Architecture | Llama-3.2-1B | Llama-3.2-1B | ResNet family |
| Number of Layers/Blocks | 16 | 16 | 16 (Blocks) |
| Hidden Dimension ($d_{model}$) | 2048 | 2048 | — |
| FFN Inner Dimension ($d_{ffn}$) | 8192 | 8192 | — |
| Number of Attention Heads | 32 | 32 | — |
| Vocabulary Size | 50,280 | 50,280 | — |
| Block Size | 2048 | 2048 | — |
| Classes | — | — | 1000 |

Table 3: **Architectural details of the primary models used in our experiments.** This table outlines the core parameters for the three main architectures analyzed to demonstrate the generality of the Gradient Fan-in Asymmetry phenomenon.

flows in a cascade, reducing with depth. The CascadeFormer is designed in harmony with this flow, tapering its width to match computational capacity to the signal's richness. This is achieved by progressively reducing the dimensions of the attention and feed-forward network (FFN) sub-layers. Based on a 16-layer Transformer baseline ($l \in \{0, 1, \ldots, 15\}$), we define a set of variants with modulated tapering intensity to explore this new design space.

| Variant Name | Tapering Target(s) | Intensity | Governing Parameters |
|---|---|---|---|
| CascadeFormer-A$_1$ | Attention Dimension | Low | $F_d = 4$ |
| CascadeFormer-A$_2$ | Attention Dimension | High | $F_d = 2$ |
| CascadeFormer-F$_1$ | FFN Dimension | Low | $S_f = 128$ |
| CascadeFormer-F$_2$ | FFN Dimension | High | $S_f = 256$ |
| CascadeFormer-C$_1$ | Attention & FFN | Low | $F_d = 4, \ S_f = 128$ |
| CascadeFormer-C$_2$ | Attention & FFN | High | $F_d = 2, \ S_f = 256$ |

Table 4: **Hyperparameter configurations for the CascadeFormer variants.** The parameters $F_d$ and $S_f$ control the tapering intensity for the attention and FFN dimensions, respectively. Lower $F_d$ and higher $S_f$ values correspond to more aggressive capacity reduction.

### A.2.2    DYNAMIC GRADIENT SCALING AS A CAUSAL INTERVENTION

Our intervention mechanism operates by dynamically rescaling gradients on a layer-wise basis during training. This ensures that the gradient magnitudes are harmonized across all layers before the optimizer step. The procedure is executed as follows:

1. **Layer-wise Norm Computation:** For each layer $i$ in the network, we aggregate all parameter gradients associated with it. We then compute the Euclidean ($L_2$) norm of these concatenated gradients, denoted as $n_i$.

2. **Target Norm Identification:** We identify the maximum gradient norm across all layers, $n_{\text{target}} = \max_i(n_i)$. This value serves as the reference magnitude for scaling.

3. **Scaling Factor Derivation:** A scaling factor $\lambda_i$ is calculated for each layer by dividing the target norm by the layer's individual norm: $\lambda_i = n_{\text{target}}/n_i$. To handle layers with zero gradients and prevent division-by-zero, $\lambda_i$ is set to $1.0$ if $n_i = 0$.

4. **In-place Gradient Application:** Finally, every parameter gradient within a given layer $i$ is multiplied in-place by its corresponding scaling factor $\lambda_i$. This operation normalizes the influence of each layer's gradient relative to the layer with the strongest signal.

### A.3    TRAINING HYPERPARAMETERS

All models were trained using the hyperparameters detailed in Table 5, tailored to their respective domains to ensure robust and competitive baseline performance.

| Hyperparameter | Transformer models | ResNet-50 |
|---|---|---|
| Dataset | pretokenized-dolma | ImageNet-1k |
| AdamW $\beta_1, \beta_2$ | (0.9, 0.95) | (0.9, 0.999) |
| Weight Decay | 0.1 | 0.05 |
| Learning Rate Schedule | Cosine w/ Warmup | Cosine w/ Warmup |
| Peak Learning Rate | 2e-4 | 5e-4 |
| Warmup Steps | 2000 | 5 Epochs |
| Batch Size | 64 | 512 |
| Epochs | 1 | 30 |
| Gradient Clipping Norm | 1.0 | - |

Table 5: **Training hyperparameters for all experiments.** This table provides the complete set of hyperparameters used for training the language and vision models, ensuring full reproducibility of our results.

### A.4 PRUNING HEURISTIC DETAILS

To provide full transparency for the pruning comparison presented in Table 1.

**Magnitude-based Heuristic Han et al. (2015)** Classical magnitude pruning removes small-magnitude weights and is a strong baseline for sparsification. In the block-level variant used here, a block's score is the aggregate of absolute parameter values across its trainable weights; for projection matrices, values are first reduced along the input dimension to obtain per-output scores, then aggregated within the block. Token embeddings and the LM head are excluded, and the final normalization is protected. Blocks are ranked by increasing score, and the $k$ lowest are pruned.

**Taylor-based Heuristic Ma et al. (2023)** The first-order criterion estimates loss increase from removing a block by accumulating, on a small calibration set, the absolute elementwise product of each parameter and its gradient. Scores are summed across batches; for projection matrices, per-output reduction precedes block-level aggregation. Token embeddings are excluded and the final normalization is protected. Blocks are ranked by increasing score, and the $k$ lowest are pruned.

**Similarity-based Heuristic. Gromov et al. (2025)** This heuristic, based on the representational similarity analysis which computes the angular distance between adjacent layer representations. Block of k layers whose removal yield the smallest change are considered less critical and are pruned.

**CFP (CascadeFlow Pruning) Heuristic.** In contrast, our proposed CFP method prunes layers with the lowest accumulated L2 gradient norms from training.

## B ADDITIONAL RESULTS AND ANALYSES

This section provides supplementary results that further validate and expand upon the findings presented in the main paper. We include additional layer importance profiles, detailed layer-wise trend plots, and a representational similarity analysis.

### B.1 GRADIENT FAN-IN DERIVATIONS

**Definition and Counting Rule.** We define gradient fan-in as the number of downstream transformation edges aggregated at a layer's input, $x_l$. This first-order proxy, counting the identity path, each downstream block's Jacobian branch, and the final head, captures the structural asymmetry driving GFA. It is a count of contributing channels, not a combinatorial path enumeration. For a standard $N$-block stack with one head, the fan-in at layer $l$ is:

$$\phi_l = (N - l) + 1, \quad \text{(functional blocks + identity path)} \tag{3}$$

**Fan-In Under Deep Supervision.** Architectures with deep supervision introduce an auxiliary loss head $\mathcal{L}_k$ at the output of each block $k$, with the total loss being $\mathcal{L}_{\text{total}} = \mathcal{L}_N + \sum_{k=0}^{N-1} \alpha_k \mathcal{L}_k$. The gradient at the input to block $l$, $g_l'$, now aggregates signals from all downstream loss functions:

$$g_l' = \frac{\partial \mathcal{L}_{\text{total}}}{\partial x_l} = \underbrace{\frac{\partial \mathcal{L}_N}{\partial x_l}}_{\text{Original Hierarchy}} + \sum_{k=l}^{N-1} \alpha_k \underbrace{\frac{\partial \mathcal{L}_k}{\partial x_l}}_{\text{Auxiliary Hierarchies}}. \tag{4}$$

Each auxiliary loss $\mathcal{L}_k$ (sourced at $x_{k+1}$) effectively creates a new residual sub-network for its gradient to traverse back to $x_l$. The fan-in from a single auxiliary loss at layer $k$ to layer $l$ is analogous to a standard network of length $(k + 1 - l)$ and is thus $(k - l) + 1 + 1 = (k - l + 2)$. Let's assume for simplicity it is attached at $x_k$ The fan-in is therefore $(k - l) + 1$. The total fan-in $\phi_l'$ is the sum of the fan-in from the original path and all new auxiliary paths originating from layers $l$ through $N - 1$:

$$\phi_l' = \phi_l + \sum_{k=l}^{N-1} ((k - l) + 1), \tag{5}$$

where $\phi_l = (N - l) + 1$ is the fan-in from the original network head. Substituting $\phi_l$ and letting $j = k - l$, the summation becomes:

$$\phi_l' = ((N - l) + 1) + \sum_{j=0}^{N-1-l} (j + 1). \tag{6}$$

Solving this arithmetic series and combining terms yields:

$$\phi_l' = ((N - l) + 1) + \frac{(N - l)(N - l + 1)}{2} = \frac{((N - l) + 1)((N - l) + 2)}{2}. \tag{7}$$

The result is the $(N - l + 1)$-th triangular number. The presence of the $(N - l)^2$ term demonstrates that deep supervision transforms the linear fan-in disparity into a quadratic one, amplifying GFA.

**Fan-In with Virtual Depth.** Consider 8 physical blocks unrolled into $N_{\text{virt}} = 16$ virtual positions with shared parameters: L1–L4 (1×), L5 (2×), L6 (3×), L7 (3×), L8 (5×). Let $V_i$ be the block at virtual position $i$ ($i = 1, \ldots, 16$). The fan-in for a physical layer $L_p$ sums over its virtual instances:

Consider 8 physical blocks unrolled into $N_{\text{virt}} = 16$ virtual positions with shared parameters

$$\text{FanIn}(L_p) = \sum_{i : V_i = L_p} (N_{\text{virt}} - i + 2). \tag{8}$$

This yields the following Standard → Virtual counts: L1: 9→18, L2: 8→17, L3: 7→16, L4: 6→15, L5: 5→27, L6: 4→33, L7: 3→24, L8: 2→20.

## B.2 Layer-wise Gradient and Importance Trends

The following figures provide a detailed visualization of the data from Figure 4.

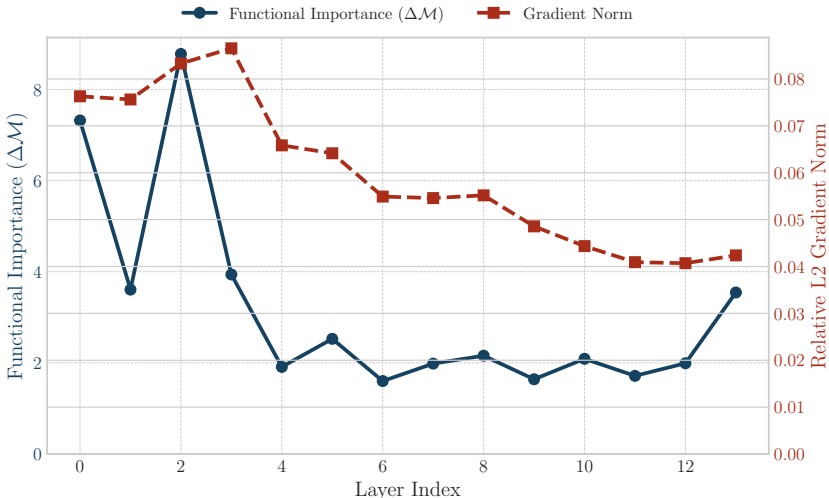

Figure 6: **Layer-wise comparison for the Vanilla Transformer.** This figure details the relationship between relative L2 gradient norm (red, dashed) and functional importance (blue, solid). The correspondence visually reinforces the correlation ($\rho = 0.62$) from Figure 4a.

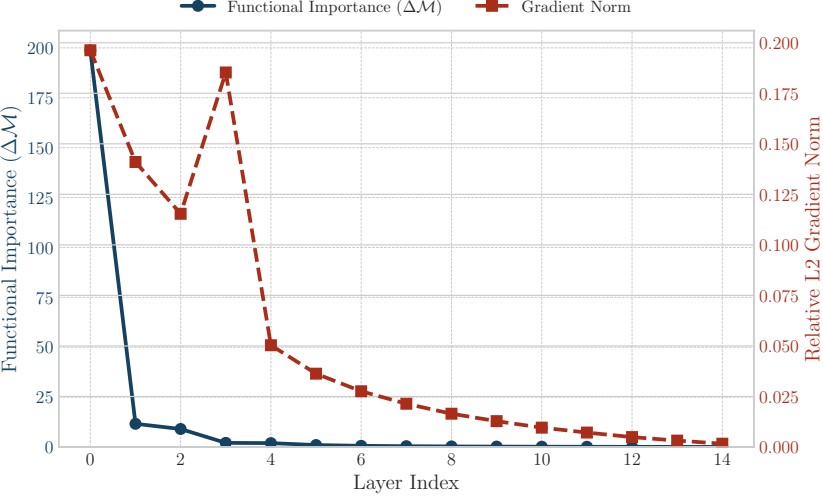

Figure 7: **Layer-wise comparison for the LayerSkip Transformer.** This figure details the relationship between relative L2 gradient norm (red, dashed) and functional importance (blue, solid). The curves track with remarkable precision, illustrating the near perfect correlation ($\rho = 0.99$) from Figure 4b.

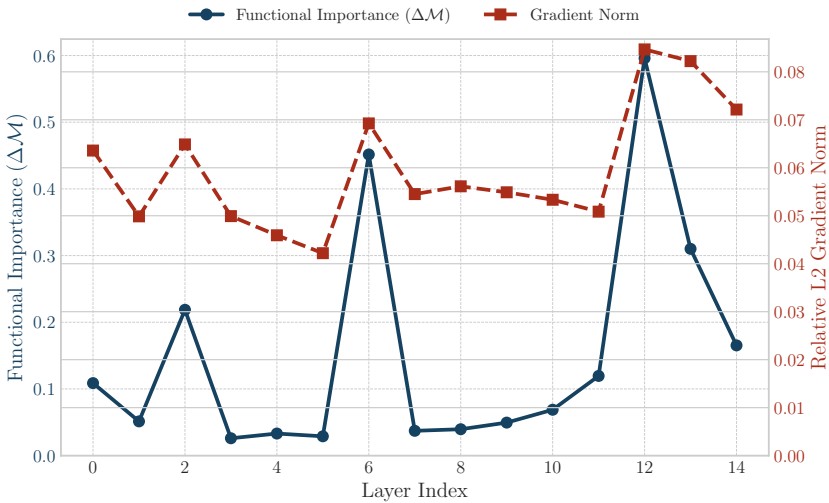

Figure 8: **Layer-wise comparison for ResNet-50.** This figure details the relationship between relative L2 gradient norm (red, dashed) and functional importance (blue, solid). A clear positive relationship is evident, corroborating the correlation ($\rho = 0.83$) from Figure 4c.

