# OpenReview forum: "Gradient Fan-in Asymmetry: The Structural Cause of Layer Redundancy in Deep Transformers"
_ICLR.cc/2026/Conference — ICLR 2026 Conference Withdrawn Submission_

### Official Review · Reviewer_Fjb4 · 2025-10-31

**Soundness:** 2
**Presentation:** 2
**Contribution:** 2
**Rating:** 2
**Confidence:** 3

**Summary:**

The paper claims to have uncovered the true reason, why deep layers add little value in transformers being Gradient-Fan-in Asymmetry.

**Strengths:**

* Better understanding transformers is a worthwhile goal is at it is the key architecture in AI!

* The authors put effort to design what they call causal interventions.

* There are also practical implications.

**Weaknesses:**

* The paper posits to have found the reasons, why deep layers add little value also supported by causal interventions. They also phrase this as a problem that needs a solution or deeper investigation. This is all good, but while the paper claims to uncover the reason, it adds little to solve the alleged problem. The paper's credibility would be much higher, if their intervention yielded some clear performance gains in terms of accuracy / perplexity. There are many mechanisms that counteract small gradients leading to small updates (e.g., using layer-wise learning rates), but none of them has yielded universal benefits. Thus, it is not even clear, if such small gradients (or the proclaimed gradient fan-in asymmetry) is an actual issue or maybe the best we can have.

* Although the authors put effort in the paper and many things are well aligned, key things are not clear and not elegant, i.e., the causal intervention and the setup appears rather convoluted (while thought through).

* Figure 3 shows large variations (e.g. for Layerskip) and increments towards the end for ResNet-50, which are against the theory. There should be more architectures / datasets being investigated to add more convidence to outputs as well as standard deviations.

* Theoretical understanding would be a plus.

Detail: Many citations have wrong bracketing, e.g., ...from OLMo OLMo et al. (2024),... -> (OLMo et al. 2024)

**Questions:**

none

---

### Official Review · Reviewer_np53 · 2025-10-31

**Soundness:** 2
**Presentation:** 2
**Contribution:** 2
**Rating:** 2
**Confidence:** 3

**Summary:**

The manuscript proposes an explanation for certain, empirically observed, aspects of deep transformer models; in particular, layer similarity/redundancy. This so called "gradient fan-in asymmetry" effect is then leveraged to: (1) compress models and (2) modify architectures.

**Strengths:**

The manuscripts hypothesis for layer redundancy seems like a plausible component/aspect of the phenomena and there is some evidence provided. Similarly, the subsequent pruning strategy and architectural modifications may be promising and seem sensible given the empirical observations.

**Weaknesses:**

The main weakness of this manuscript is, in essence, that it is spread too thin over too many ideas. Conceptually, one could see this as 2 or 2.5 manuscripts worth of topics (though certainly not content at present). There are also some aspects of the presentation that need to be (significantly) improved.

Considering the first point, the manuscript seems to try and both identify a structural explanation for layer redundancy and turn that observation into a pruning strategy. However, in doing both I think that neither piece is done sufficiently well. There is some empirical evidence provided for the hypothesis, but it is not broad or rigorous enough to feel like it is strong justification—a strong manuscript on just the first point above would need a more systematic treatment of the experiments (e.g., over more varied model sizes, training parameters, data sets, etc.) to bolster its claim.

Similarly, the part of the manuscript on pruning/compression does not do a good enough job with that problem (which is a big one on its own). A stronger manuscript in that direction. A stronger manuscript in that direction needs a more comprehensive set of experiments (e.g., comparisons with methods such as [1] and [2]; see also references therein) and better engagement with the broader compression literature (e.g., trade offs with structure pruning methods).


[1] Lin, Sihao, Pumeng Lyu, Dongrui Liu, Tao Tang, Xiaodan Liang, Andy Song, and Xiaojun Chang. "Mlp can be a good transformer learner." In Proceedings of the IEEE/CVF Conference on Computer Vision and Pattern Recognition, pp. 19489-19498. 2024.

[2] Zhang, Hanxiao, Yifan Zhou, and Guo-Hua Wang. "Dense vision transformer compression with few samples." In Proceedings of the IEEE/CVF conference on computer vision and pattern recognition, pp. 15825-15834. 2024.

Compounding these issues is that the manuscripts presentation needs improvement. Some of this is simply grammatical/stylistic (e.g., there is significant redundancy in certain ideas such as how many times the observation is noted to be structural). But the larger issue is a lack of precision. For example, the latter two parts of the manuscript on pruning and the cascade former have components to the methods/architecture that are described in prose and could be much more formally (and, therefore, unambiguously) stated. Similarly, the early part of the manuscript lacks a crisp, mathematical statement of the hypothesis (the bits are all there but they are somewhat diffuse) and that, actually, makes it somewhat challenging to assess the strength of support for the claim via proxy measurements.

I think there is the possibility for a decent contribution here, but feel that the manuscript needs to lean into one of its two objectives and consolidate its narrative around that (with appropriate additions/revisions).

**Questions:**

- What exactly is being plotted in Fig. 1(a)? The description is not precise enough.

---

### Official Review · Reviewer_AByr · 2025-11-01

**Soundness:** 2
**Presentation:** 2
**Contribution:** 2
**Rating:** 4
**Confidence:** 4

**Summary:**

This paper identifies "Gradient Fan-in asymmetry", where the structure of pre-layernorm architectures means that early layers have higher gradient norms that later layers because they have more "downstream functional paths", as a structural driver of late layer redundancy in deep transformers. The authors demonstrate this through two causal interventions: equalising per-layer gradient norms across layers, and repeating late layers to increase the number of downstream functional paths for late layers. Finally, the authors propose two practical insights motivated by GFA: a pruning method and a modified architecture that reduces width of the hidden dimension (mlp dimension or number of heads) in resblocks to account for GFA and save compute in late layers that have more redundant gradients.

**Strengths:**

- The topic of studying layerwise training dynamics is an interesting one, and the paper proposes some interesting insights and experimental setups to that end, like the gradient rescaling in section 4.4.
- The authors motivate some practical applications of their proposed GFA, like the layerwise pruning strategy, which demonstrate some initial promising results.

**Weaknesses:**

My main concern is that while the paper touches on an interesting topic, I do not feel like it has been properly and thoroughly developed at present, namely:
- A few of the experimental results are not convincing. For example in figure 3 the resnet 50 does not seem to have an obviously decreasing layerwise grad norm as a function of layer, and the Vanilla Transformer is also quite a weak signal. The claim "the Vanilla Transformer closely tracks its theoretical fan-in" seems to be a stretch looking at the figure, which suggests maybe the theory is not quite as developed as one might hope. In any case, I would like to see error bars over seeds for the empirical runs ideally.
- Likewise the fits on figure 4 do not appear so strong, especially for the layerskip transformer where it just seems like there is a big outlier layer for functional importance?
- In section 4.4, which optimiser is used for the "equalizing magnitude does not restore importance" experiment? Ideally we would see at least 1 optimiser that is theoretically "scale-invariant" like Adam, and then one that is not, like SGD. The resnet50 setting should work for SGD if the text transformers do not train well with SGD. This feels important because as the authors write on line 161: "An optimizer like AdamW can rescale this gradient...". Moreover, how does the resulting gradient rescaled model perform? On the topic of section 4.4, again in Figure 5 it seems like the vanilla model has a dominant first layer, rather than the premise that "deep layers are redundant" on which the paper is motivated.
- Personally, I found the more exploratory analysis up to and including sections 4.4 more interesting than the applications section, which I also feel was a bit undercooked: for example, finetuning results feel important to make a more pressing claim for the practicality of CFP.

**Questions:**

- Another interesting experiment for section 4.4 would be to apply a residual scaling e.g. https://arxiv.org/abs/2002.10444 in order to equalise the gradient norms to the weights in different layers, as an alternative architectural intervention besides repeat layers. I would be curious how this performs (there is a reason we want to downweight layers for signal propagation/preserving the identity stream). Also curious how this gradient norm/functional importance would look for a post-LN model.
- Have the authors investigated how this behaviour scales? If it is too expensive to run bigger models than the ones presented in the paper, they could also see the behaviour at smaller scales. In this case, the evolution throughout training seems like an interesting question to explore (for fixed model size), but also changing to smaller model sizes could be worthwhile.

---

### Note · Authors · 2025-11-19

I have read and agree with the venue's withdrawal policy on behalf of myself and my co-authors.